# The Volume-Regulated Anion Channel in Glioblastoma

**DOI:** 10.3390/cancers11030307

**Published:** 2019-03-05

**Authors:** Martino Caramia, Luigi Sforna, Fabio Franciolini, Luigi Catacuzzeno

**Affiliations:** 1Department of Chemistry, Biology and Biotechnology, University of Perugia, Perugia 06123, Italy; fabio.franciolini@unipg.it; 2Department of Experimental Medicine, University of Perugia, Perugia 06132, Italy; luigi.sforna@unipg.it

**Keywords:** GBM, VRAC, IClswell, hypoxia, microenvironment

## Abstract

Malignancy of glioblastoma multiforme (GBM), the most common and aggressive form of human brain tumor, strongly depends on its enhanced cell invasion and death evasion which make surgery and accompanying therapies highly ineffective. Several ion channels that regulate membrane potential, cytosolic Ca^2+^ concentration and cell volume in GBM cells play significant roles in sustaining these processes. Among them, the volume-regulated anion channel (VRAC), which mediates the swelling-activated chloride current (IClswell) and is highly expressed in GBM cells, arguably plays a major role. VRAC is primarily involved in reestablishing the original cell volume that may be lost under several physiopathological conditions, but also in sustaining the shape and cell volume changes needed for cell migration and proliferation. While experimentally VRAC is activated by exposing cells to hypotonic solutions that cause the increase of cell volume, in vivo it is thought to be controlled by several different stimuli and modulators. In this review we focus on our recent work showing that two conditions normally occurring in pathological GBM tissues, namely high serum levels and severe hypoxia, were both able to activate VRAC, and their activation was found to promote cell migration and resistance to cell death, both features enhancing GBM malignancy. Also, the fact that the signal transduction pathway leading to VRAC activation appears to involve GBM specific intracellular components, such as diacylglicerol kinase and phosphatidic acid, reportedly not involved in the activation of VRAC in healthy tissues, is a relevant finding. Based on these observations and the impact of VRAC in the physiopathology of GBM, targeting this channel or its intracellular regulators may represent an effective strategy to contrast this lethal tumor.

## 1. The Glioblastoma

Glioblastoma (GBM) is the most common and aggressive primary brain tumor [1,2]. It normally results in the patient death within 15 months, despite modern diagnostics and treatments [3,4]. The main obstacles in treating GBM remain its high migratory and invasive potential into healthy brain parenchyma [5,6,7] that prevents complete surgical removal of tumor cells, and its resistance to chemotherapy, that makes pharmacological treatments less effective [2,8]. As a result, the tumor cells generate new foci distant from the main tumor mass, which are by and large the primary cause of mortality (>90%) in GBM patients. 

GBM displays a high level of heterogeneity. Recent advances in molecular genetics have shown that the previous categories into which gliomas were grouped, based on their histological features, correlate surprisingly well with only a small number of frequently occurring molecular traits. In accordance with these observations and the 2016 revised criteria of the World Health Organization, gliomas can now be subdivided into three major groups based on only two parameters: the mutation status of isocitrate dehydrogenase 1 (IDH1, a protein mutated in most low-grade infiltrating gliomas, and easily detected by immunohistochemical staining) and the deletion status of chromosome arms 1p and 19q. The three groups are: (i) IDH-wild type (astrocytoma); (ii) IDH–mutant (astrocytoma); (iii) IDH-mutant and co-deletion of 1p/19q (oligodendroglioma). The case of 1p/19q co-deletion and IDH-wild type is too rare (occurrence less than 0.1%) to make a separate group. Astrocytoma and oligodendroglioma not falling into these three groups enter into the ‘‘not otherwise specified’’ (NOS) category [9]. Notably, each of these groups can further carry other somatic alterations in different loci. For instance, the IDH-wild type astrocytomas and GBM often show remarkably high levels of EGFR, whose activation by EGF induces uncontrolled tumor cell proliferation, while the IDH-mutant astrocytomas display high rate of mutations in the tumor suppressor gene TP53 [10]. 

As already stressed, the poor benefit of both surgery and chemotherapy critically derives from the high migratory and invasive potential of GBM into healthy brain [5,6,7]. GBM cell migration is a highly regulated multistep process, whereby tumor cells escape the primary tumor mass and form metastatic new foci in non-tumor brain tissue at new distant locations. The metastatic process starts with GBM cells losing adhesion with surrounding elements, avoiding the cell death often associated with extracellular matrix (ECM) disconnection, and acquiring a highly migratory phenotype [11,12], which is a major component of the metastatic cascade. Cell migration can be viewed as a repeated cycle of protrusion of the leading edge—with formation of lamellipodia, filopodia, invadopodia—and retraction of the cell rear [13]. Cells repeatedly go through this cycle by changing their morphology and size. The acquisition of a fusiform shape and the changes of volume are instrumental to invading cells in order to make their way through the narrow spaces of surrounding parenchyma [13]. 

The basic mechanisms underlying migration of GBM cells are equivalent to those present in most types of migratory cells; to the point that, as migration is a property of many non-tumor cells, although restricted to specific developmental stages or environmental conditions, the migration of tumor cells is often viewed as the result of dysregulation of specific biochemical pathways as, for example, PI3K/Akt/mTor [14,15] that in non-pathological conditions tightly control the migration of cells. Alterations in PI3K signaling in GBM are mainly related to somatic mutations of the p110α catalytic subunit, which have been found in 15% of GBM (both adult and pediatric) patients [16]. Downregulation of p110α expression by RNA significantly inhibited migration and invasion in GBM cells [17,18]. PI3K/Akt/mTor pathway was also reported to be activated by hypoxia, and in this activated form it enhanced migration and invasion of GBM U87 cells. Conversely, migration and invasion were suppressed when either PI3K/Akt/mTor pathway or HIF-1α was inhibited by the specific siRNA [19]. The PI3Ks family is also involved in the control of cell adhesion and actin reorganization during cell migration [20,21], and its aberrant activation in GBM is associated with increasing tumor grade and with high motility and invasion rates [22,23].

## 2. The Swelling Activated Chloride Current

### 2.1. General

Chloride channels have classically represented a poorly investigated class of ion channels. While Na and K channels were recognized as determinant for critical processes such as the action potential generation and cell excitability in general, and Ca channels were considered pivotal in synaptic release and muscle contraction, Cl channels were assigned no specific role in major biological processes. On the contrary, their currents were viewed by the electrophysiologists of the ‘60s and the ‘70s just as a hindrance in their experiments (the painful “leak current”). In part this view originated from the evidence that unlike the Na, K and Ca channels, whose studies could greatly benefit from the availability of very effective drugs and toxins, no such tools were available (and still today they lag much behind) for Cl channels, with the result that their functional and molecular characterization, as well as specific functional tests have been unpractical for decades.

Several families of Cl channels were later identified, and are known today to underlie a large range of biological functions, including epithelial fluid secretion, cell volume regulation, pH control of cytoplasm and cellular organelles. Mainly based on the type of stimulus that activates them, together with biophysical and pharmacological properties such as voltage dependence, permeability sequence, single-channel conductance, and sensitivity to toxins or synthetic modulators [24], Cl channels have been grouped into five main families: (i) neurotransmitter-gated Cl channel receptors (GABA_A_R and GlyR); (ii) voltage-gated Cl channels (ClC-family); (iii) cystic fibrosis transmembrane conductance regulator (CFTR); cAMP-PKA activated Cl channel; (iv) Ca^2+^-activated Cl channels (CaCCs); (v) voltage regulated anion channels (VRAC).

In this review, we will focus on the last group of Cl channels, VRAC. These channels are expressed in virtually all cells [25,26,27,28,29,30,31], including GBM [11,32,33,34] and related astroglial cells [35]. In GBM, VRAC participates in enhancing tumor malignancy [36,37]. We will first provide a brief introduction on the salient properties of the channel and its recently defined molecular identity, and then address the conditions GBM experiences as a result of the blood–brain barrier (BBB) deterioration especially in the highly neovascularized regions where GBM cells may easily come in contact with blood serum [38]. We will finally tackle the mechanisms controlling the hypoxia-induced GBM malignancy, which assign VRAC a leading role. 

### 2.2. Basic Biophysical Properties of VRAC

Typically, VRAC are experimentally activated by cell swelling resulting from exposure to extracellular hypotonic solutions (Figure 1A,B,D). The resulting IClswell shows weak outward rectification (higher conductance at positive potentials—from 10–20 pS at negative potentials to 50–80 pS at positive potentials; Figure 1B,D), and variable time- and voltage-dependent inactivation (Figure 1D), whose rate is strictly related to pH, extracellular Mg^2+^ and Cl^−^. These features are, however, highly variable among cell types [27,30,31,32,39,40,41]. 

VRAC displays broad permeability to several anions, with the following sequence: SCN^−^ > I^−^ > NO_3_^−^ > Br^−^ > Cl^−^ > HCO_3_^−^ > glycine > F^−^ [30,31,39]. This is an Eisenman type 1 halide permeability sequence (I^−^ > Br^−^ > Cl^−^ > F^−^) corresponding to an anion binding site of weak field strength. VRAC is also permeant to neurotransmitters (glycine, glutamate, ATP) and other signaling molecules, suggesting that it might have a role in paracrine or autocrine signaling [42,43,44]. Fitting the relative permeabilities of these ions to their Stoke’s diameter, VRAC resulted to have a pore diameter of about 11 Å. Better estimates using 4-sulfonic-calix(n)arene as permeation reporter—namely the observation that calix(4)arene but not calix(6)arene permeated the channel—led to the conclusion that VRAC pore diameter was between 11 and 17 Å, with a most likely value of 12.6 Å [45,46]. These values are compatible with the release of organic osmolytes like taurine and glutamate, and also with the uptake of cisplatin and carboplatin [47] which have maximal diameters between 3.0 and 6.0 Å. More recently significantly smaller pore dimensions have been derived from high-resolution structures obtained with cryo-EM and X-ray crystallography of homo-exameric LRRC8A channels (diameters lower than 6 Å [48,49]). These small pore diameters are however somehow expected from the absence of LRRC8D subunits in these constructs, which have been shown to form wider pores and confer broader substrate specificity [47,50].

Initial studies of native, elementary IClswell estimated a single-channel slope conductance of 50–80 pS at positive potentials [50,51,52]. These data were confirmed by recent tests carried out after the molecular identification of VRAC. Varying LRRC8 isoforms, VRAC reconstituted in lipid bilayers gave a single-channel conductance ranging from 10 to 50 pS at −100 mV, when exposed to a hypotonic solution [53].

### 2.3. Pharmacology of VRAC

One major problem in studying VRAC, and one of the reasons why it took so long to identify its molecular counterparts, is the lack of selective channel blockers. There are several nonspecific agents that somehow discriminate among Cl channels, as DIDS (4,4′–diisothiocyano-2,2′-stilbenedisulfonic acid) NPPB (5-nitro-2-(3-phenylpropylamino)benzoic acid), DCPIB (4-(2-butyl-6,7-dichloro-2-cyclopentylindan-1-on-5-yl)oxybutyric acid), tamoxifen, niflumic acid , which at micromolar concentrations inhibit VRAC (Figure 1A–D). A more selective antagonist and, as of today, widely used inhibitor of VRAC is DCPIB [54]. On this ground DCPIB has been used to probe the role of VRAC in GBM [36,55,56], in spite of its off-target blockade/activation of inward rectifier (Kir) and TREK K channels, respectively [57,58]. Most recently a 3D cryo-EM structure of the DCPIB-inhibited VRAC LRRC8A was reported, which showed that DCPIB blocks VRAC by plugging the external mouth of the channel. The interaction between the blocker and the channel occurs electrostatically, with the DCPIB carboxylic acid group interacting with the arginine at position 103 (R103). This position is the narrowest of VRAC pore, is located in the external portion of the channel, and represents the selectivity filter. Mutation of R103 to phenylalanine or alanine decreases the blocking power of DCPIB, and increases the relative Na^+^ permeability, respectively, of LRRC8A/8C heteromeric channels [48,59].

DCPIB was also reported to exert neuroprotective effects supposedly caused by the inhibition of glutamate release via VRAC [60,61]. An excessive increase of extracellular glutamate would lead to the activation of Ca^2+^-permeable NMDA glutamate receptors, which results in cytosolic Ca^2+^ overload and neuronal death [62]. DCPIB strongly inhibits also the glial glutamate transporter GLT-1 (important for maintaining low extracellular glutamate levels [63]), connexin hemichannels, and Ca^2+^-activated Cl channels [64]. Carbenoxolone (CBX), a known neuroprotective agent and blocker of gap junction channels formed by connexins, was also found to effectively inhibit VRAC in cultured rat cortical astroglia (IC_50_ ~15 μM and a Hill coefficient of ~2 [65]). Under voltage-clamp, CBX inhibited IClswell when applied from the extracellular side. However, full block, as well as the washout of CBX, were both rather slow, and the possibility that the relatively hydrophobic molecule crosses the membrane blocking from the inside, could not be excluded.

### 2.4. Activation of VRAC

The nature of cell volume signals and downstream pathways that link cell volume increase to VRAC activation still remains a largely open question. Several intracellular molecules and signaling pathways have been proposed, but none has proved crucial enough, across a significant number of different cell types, to gain a general stand. VRAC is generally taken to be activated by cell swelling, which is attained experimentally by perfusing cells with hypotonic solutions. However, the Nilius group and others showed that it is not cell swelling *per se* (i.e., mechanical stress on the membrane) that activates the channel, but rather a decrease of the intracellular ionic strength [66,67,68]. Many studies report the involvement of various transduction pathways in the activation of VRAC, that require intracellular ATP, threshold cytoplasmic Ca^2+^ levels (secondary to purinergic or bradykinin receptors stimulation), increased reactive oxygen species (ROS), activation of G protein-dependent Rho and ERK pathways, phosphorylation cascades, or the specific polymerization status of actin filaments. Most of these mechanisms and cascades have been reviewed extensively (see for instance [69]). A clear unifying mechanism accounting for the major available data is, however, not yet available.

In human GBM cells, we recently reported a peculiar transduction pathway, which was never described in non-tumoral cells, that mediates IClswell activation [33]. We found that IClswell activation by hypotonic swelling requires the activity of a U73122-sensitive phospholipase C (PLC), and that the membrane-permeable diacylglycerol (DAG) analog OAG was able to activate IClswell. We further found that R59022, an inhibitor of DAG kinase (DKG), reverted IClswell activation, suggesting the involvement of phosphatidic acid (PA). IClswell activation also required the activity of a EHT1864-sensitive Rac1 small GTPase and the consequent actin polymerization, as IClswell activation was prevented by cytochalasin B. Therefore, two signaling pathways seem to participate in the activation of IClswell by hypotonic stress, namely the PLC/DAG/DGK/PA pathway and the Rac-mediated cytoskeleton remodelling [70]. What remains to be established is whether they represent two separate pathways that work in parallel, but converge to VRAC, or if they form a sequentially coupled linear pathway that links together the two modulatory segments, as sketched in Figure 2. In support of this latter view is the observation made in varying cell models that PA activates Rac and induces polymerization of actin filaments, an event known to be involved in VRAC activation [71,72,73,74]. 

In Figure 2 we present the hypothetical sequentially-coupled linear pathway leading to VRAC activation in GBM cells following hypotonic stress. According to this model, cell swelling leads to the activation of PLC and the production of DAG, which is phosphorylated to PA by DGK. PA would then activate Rac, which in turn stimulates the polymerization of actin filaments, and ultimately activates VRAC. We stress once again that this scheme depicts just one of the possible pathways involved in VRAC activation, and the issue requires further investigation.

### 2.5. Molecular Identity of VRAC

In 2014, thirty years after its first functional reports, and a long history of false positive candidates [29], two studies simultaneously identified the leucine-rich repeat eight heteromers (LRRC8A-E) as essential components of VRAC [75,76]. VRAC was found to require an obligatory LRRC8A subunit (its knockdown or deletion abolished IClswell, and it could not be replaced by any of the other family members), and at least one additional LRRC8B-E isoform, to be functional [76].

All LRRC8A-E isoforms are made of four transmembrane segments and a leucine-rich repeat domain containing up to 17 leucine-rich repeats at the C-terminal segment, and the functional VRACs are thought to be organized in a hexameric structure, similar to the pannexin family of ion channels with which LRRC8A-E share high homology [77] (Figure 3a,b ). The many possible combinations of LRRC8 heteromers for forming the channel, may account for the different properties of VRACs, such as the permeability to ions and organic compounds, inactivation rate, single channel conductance, blocker sensitivity, which have been reported by electrophysiological and pharmacological studies over the years. No specific role could be conclusively assigned to any of them, except for the LRRC8D found to be critical for the cisplatin and carboplatin uptake inside the cell, and most likely involved with drug resistance (to be discussed below in Section 2.7). 

However, major questions regarding VRAC function were raised right after the identification of VRAC molecular counterparts, when it was observed that several cell lines could survive and proliferate in the absence of all the five isoforms LRRC8A-E [76,77,78], and LRRC8A isoform was not required for generating IClswell [79,80]. Not only have these observations raised questions regarding essential/non-essential isoforms for making functional VRAC; they question more directly the largely assumed crucial role of VRAC in many critical cell functions, including cell volume regulation, proliferation, migration, apoptosis, and implicitly suggest the presence of redundant VRAC-independent mechanisms capable of sustaining volume regulation or cell migration. The question is however still open on whether VRAC expressed in specific cells may have other as yet unidentified molecular counterparts.

### 2.6. VRAC Classical Role: Control of Cell Volume

The activation of VRAC by cell swelling suggests that its primary physiological role is associated to cell volume control. VRAC are ubiquitously expressed in mammalian cells, and their activation is experimentally induced by cell swelling, resulting from cell exposure to external hypotonic solution. In response to swelling, cells initiate a complex homeostatic process mainly consisting in the activation of VRAC together with K conductances, resulting in a net efflux of Cl^−^ and K^+^ ions (a Cl^−^ efflux can take place in GBM cells as they actively accumulate intracellular Cl^−^ ions well above the Nernst equilibrium concentration). This is followed by osmotic water loss and reestablishment of the original cell volume, a process known as regulatory volume decrease (RVD). As expected, blockade of VRAC impairs the RVD process and the restoration of the physiological volume of the cell. 

It is well known that key cellular activities such as proliferation, migration and programmed death require significant changes in cell volume, and are thus sensitive to those processes underlying cell volume regulation. For example cell proliferation is stimulated by cell swelling and inhibited by cell shrinkage [81,82,83,84] while programmed cell death is associated to a significant cell volume shrinkage [85]. The migratory process is instead associated with cycling changes in cell volume that lead to cell front advances and cell rear retractions [86]. It is thus not surprising that VRAC and associated IClswell are critically important in all those cellular processes that involve a change of cell volume and shape, such as cell proliferation, migration and death (besides those processes and functions dependent on the osmolyte transport through the membrane, such as setting the cell resting potential or intracellular pH). In GBM, IClswell has been suggested to facilitate cell infiltration through the narrow spaces of the brain parenchyma, where major changes of cell volume and shape are required [11,87,88,89,90,91,92]. 

### 2.7. VRAC in Drug Transport and Drug Resistance

The identification of the molecular counterpart of VRAC has helped greatly to clarify the relation between VRAC and cell drug resistance. It has long been known that dysfunction, downregulation or block of VRAC are associated with resistance to chemotherapics (e.g., platinum-based compounds), and that cisplatin [38] cytotoxicity parallels with VRAC activity [56,93,94,95,96]. For instance, in KCP-4 epidermoid cancer cell line displaying strong resistance to cisplatin, VRAC activity was found to be essentially absent; cisplatin resistance could however be attenuated by partially restoring VRAC activity [95]. In several cell lines it was found that cisplatin activated caspase-3 and induced apoptosis were more effective in hypotonic conditions, when VRACs are open. Altogether these observations pointed to VRAC mediating cisplatin uptake [96]. 

Following the identification of LRRC8 protein family as the molecular constituents of VRAC, it was not only conclusively established that VRAC transported cisplatin and carboplatin inside the cell, but also that the efficiency of transport depended on the composition of the LRRC8 subunits [47,97,98]. Planells-Cases et al. found that the lack of either LRRC8A or LRRC8D subunit increased the cell resistance to apoptosis upon cisplatin or carboplatin treatment, suggesting that both subunits were needed for cisplatin transport inside the cell [48]. Lack of LRRC8A-dependent drug resistance could easily be understood on the grounds that LRRC8A is the obligatory subunit required for membrane insertion of the channel [75,76]. A different scenario formed for LRRC8D, a dispensable subunit, whose loss did not decrease IClswell, but inhibited markedly the RVD upon hypotonic stress, and cisplatin-induced apoptosis.

These effects were explained when it was found that VRAC made of LRRC8A and LRRC8B/C/E, but not LRRC8D, drastically reduced the swelling-induced efflux of taurine as well as influx of cisplatin (cf. Figure 4). Since taurine is highly present in the cytoplasm (10–50 mM) and greatly contributes to the osmolite efflux in the RVD process, and cisplatin is a strong apoptotic inductor, it becomes clear why the lack of LRRC8D subunit that heavily prevents taurine efflux and cisplatin uptake resulted in a marked inhibition of RVD and cisplatin-induced apoptosis upon hypotonic stress.

## 3. Activation of VRAC by Pathologically Relevant Conditions

Two among the main indicators of GBM malignancy are enhanced migration/invasion potential into brain parenchyma, and escape from cell death. VRAC activity and related IClswell are key determinants in promoting and sustaining both processes. IClswell is pivotal as it provides, in conjunction with K currents, the ionic fluxes and consequent osmotic water movement needed for GBM cells volume changes, which is a precondition for both cell migration/invasion and cell death. 

The main point to be addressed remains however how IClswell is activated in a proper physiopathological setting where GBM cells hardly experience hypotonic conditions such as those needed to activate IClswell in vitro. The activation of this current in GBM patients’ brain must, then, take place via other signal pathways and mediators. We here discuss two of them that may have major physiopathological relevance: blood serum and hypoxia.

### 3.1. Blood Serum

Under isotonic conditions IClswell has been shown to be activated by fetal calf serum (FCS). The application of isotonic solutions containing 1–10% FCS activates a current with features that are congruent to IClswell activated by hypotonic conditions (FCS-activated current had a reversal potential close to ECl, at potentials above −60 mV showed a marked voltage-dependent inactivation, and was effectively inhibited by 100 µM NPPB and 500 µM DIDS [32,92]). Further evidence for the consonance between the two currents is the observation that FCS activated the current using several transduction molecules previously found to be involved in IClswell activation upon exposure to hypotonic stress, such as PLC and DGK (FCS-activated current could be inhibited by the PLC and DGKs specific inhibitors U73122 and R59022, respectively [33]; cf. Figure 2). Moreover, the amount of block of the FCS-activated current by NPPB and DIDS was comparable to that found for IClswell stimulated by hypotonicity in the same glioblastoma cell types. Finally, the FCS-activated current was mostly reverted by a 30% hypertonic solution [33]. All these data strongly indicate that the current activated by FCS is the IClswell activated by hypotonic stress.

These observations may have substantial relevance on account that in vivo GBM cells hardly experience hypotonic conditions such as those required to activate IClswell. Blood serum could instead be the activator of IClswell, given that brain tumors are known to induce partial breakdown of the BBB and increase vascular permeability and serum leakage into the surrounding brain tissue. The vasculature of a healthy brain is entirely surrounded by astrocyte endfeet which, together with the endothelial cell layer and basement membrane, form the BBB that strictly controls the passage of ions, molecules and cells between blood serum and the brain [99]. The endothelial cell layer and brain tissue are not tightly sealed all over the entire interface, but leave spaces—the Virchow-Robin spaces—which invading GBM cells can use to move forward and invade new brain areas (whether glioma cells preferentially infiltrate the perivascular space in response to environmental cues or because it represents the least resistant path is not clear). When the path is blocked GBM cells rip the astrocyte endfeet off the endothelial layer to make room for their passage, and consequently disrupt the integrity of the BBB.

In a recent study to test whether FCS would increase the invasive potential of GBM cells, primary GBM cell lines were isolated and cultured either in a medium containing bFGF and EGF but no FCS, or in a medium supplemented with 10% FCS [100]. The study found that FCS-cultured GBM cells enhanced migration and invasion power in vitro (wound healing assay) as compared to zero FCS-cultured cells. Furthermore, the tumors originating from intracranial implanted serum-cultured cells showed higher migration/invasion than tumors derived from zero FCS-cultured cells [100]. Given that zero FCS/bFGF+EGF medium is known to enrich for GBM stem cells while FCS promotes cell differentiation, these results suggest that GBM stem cells behave as the neural stem cells residing in the subventricular area, which remain in situ, while neuroblasts move towards the olfactory bulb. In other words, they are not the stem cells that migrate through the brain, but the more differentiated neuroblast progenitors [101,102].

Unfortunately, the FCS component/s responsible for the observed effects on IClswell is presently unknown. Notably, several growth factors present in FCS at relatively high concentration, such as the fibroblast growth factor (FGF), the platelet-derived growth factor (PDGF) and insulin were all able to increase (Ca^2+^)i in U87-MG cells, although not as much as those produced by FCS [103]. A number of other molecules, potentially able to activate PLC and present in FCS, have been shown to have membrane receptors on GBM cells, such as acetylcholine [104], LPA [105], bradykinin [105], ATP and glutamate [106].

### 3.2. Hypoxia

Hypoxic environments drive cells to switch to an anaerobic metabolism, whereby glucose is converted to pyruvate instead of entering the TCA cycle, and then reduced to lactate. Inside the cell, the overall equilibrium of this reaction strongly favors lactate formation, with a ratio of about 10–25 lactate to 1 pyruvate molecules, which makes lactate the actual end product of the anaerobic glucose degradation, and the excessive accumulation of hydrogen ions (low pH), the final condition of the cell (Figure 5a). This condition is also known as lactic acidosis. Hypoxia and ensuing anaerobic metabolism also lead to decreased ATP synthesis and consequent ATP depletion. 

ATP is required for many processes within the cell, including active membrane transport, most relevant here. Depletion of ATP to less than 10% of normal levels, which may easily occur even under moderate, continued hypoxia, has compelling effects on many critical cellular processes such as the activity of the ATP-dependent Na^+^/K^+^ pump. The inability of ATP-dependent pumps to maintain the homeostasis of the intracellular environment, broken by external Na^+^ drifting towards its thermodynamic equilibrium, represents the primary cause of hypoxia-induced cell death [107,108]. Na^+^ accumulation inside the cell also brings osmotic water inside, causing cell swelling (Figure 5a). In neuronal and glial cells, hypoxia-dependent lactic acidosis and ensuing cell swelling, often lead to necrotic cell death, as these tissues under hypoxic conditions are not capable to efficiently activate the RVD process [109,110,111]. In addition, Na^+^ accumulation inside the cell, also leads to membrane depolarization and increased Ca^2+^ influx through voltage-gated Ca channels. The rise of intracellular Ca^2+^ concentration activates Ca^2+^-dependent phospholipases and proteases that lead to cellular swelling and ultimately to necrotic cell death [112].

Contrary to mature astrocytes that rapidly die under hypoxia [113], GBM cells are able to survive to similar insults [114], and this resistance to hypoxia-induced death is certainly functional to let tumor cells survive under hypoxic conditions and acquire a more aggressive phenotype. Indeed, increasing GBM malignancy strongly correlates with insufficient blood supply and hypoxic level. Many hypoxic regions are found in GBM, including a central large necrotic core and multiple thrombotic foci surrounded by pseudopalisading cells that migrate away towards more oxygenated areas [115,116]. Hypoxia represents a major driving towards tumor malignancy, which is gained through increased invasion, resistance to apoptosis, chemo and radioresistance, and tumorigenic cancer stem cells development [7,117]. While in most cases these malignant phenotypes are gained as a result of hypoxic conditions acting on gene expression via hypoxia-induced factors (HIFs [117]), in few instances they are mediated by hypoxia acting acutely and directly on minimal targets. Here we describe one such case involving VRAC, its high expression in GBM, and how its activation by hypoxia protects GBM cells from hypoxia-induced death.

We recently found that hypoxic conditions (attained by acute application of hypoxic solution to GBM cell lines) activate a Cl current with major biophysical and pharmacological features of IClswell [34]. Hypoxic conditions also increased the cell volume of GBM cells (by approximately 20%), possibly as a result of hypoxia-induced accumulation of lactate and Na^+^ ions inside the cell, with consequent water influx and cell swelling. We also found that the cell volume increase activated the regulatory volume decrease (RVD) process, mediated mainly by VRAC. Knowing that sustained hypoxia-induced cell swelling is usually regarded as a death insult, we guessed that the hypoxia-activated VRAC in GBM cells would represent a survival reaction to limit cell swelling and prevent necrotic death, which often occur under hypoxic conditions (Figure 5b). We tested this idea and found that the specific VRAC blocker DCPIB inhibited significantly the RVD process, and most importantly it sensibly increased the hypoxia-induced necrotic death in these cells [34]. Taken together, these results suggest that VRAC is involved in the survival of GBM cells in a hypoxic environment, and significantly contributes to their malignant phenotype. These observations indicate that VRAC inhibition may represent an effective therapeutic target for GBM. 

In many different preparations, including normal astrocytes, VRAC activity has been shown to require non-hydrolytic binding of intracellular ATP [25,118,119,120]. Notably, in GBM cells VRAC activation does not appear to depend on ATP. In human glioma cells VRAC currents could be activated upon cell swelling also in the absence of intracellular ATP, and addition of ATP did not appreciably affect VRAC current [89]. This finding is consistent with our unpublished observations that in GBM cells VRAC is strongly activated by hypoxia, a condition that brings to generalized metabolic inhibition and ATP depletion [121]. It needs however be recalled that in a condition of depleted ATP, the ensuing reduced activity of ATP-dependent Na^+^/K^+^ pumps will progressively dissipate the K^+^ gradient and the resulting K^+^ efflux. This occurrence will in turn inhibit the electrically coupled Cl^−^ efflux through VRAC, and the very ability of VRAC to reduce the cell volume. 

## 4. Conclusions and Perspectives

We have shown that VRAC are expressed in GBM cells, and can be activated by two conditions most frequently found in pathological GBM tissues, namely high serum and severe hypoxia. Upon their activation, that likely occurs in a typical GBM microenvironment, these channels will stimulate cell migration and evasion from cell death, features that increase the malignancy of this tumor. Our finding that the signal transduction pathway leading to VRAC activation in GBM cells appears to involve GBM specific intracellular components, such as DGK and PA, apparently not involved in the activation of VRAC in healthy tissues, is also relevant. It remains to be demonstrated whether our observations made in GBM cell lines apply to GBM cells in situ. Based on these observations the inhibition of this channel or its intracellular positive regulators is predicted to contrast malignancy of GBM cells. With the recent discovery of the molecular nature of these channels, new selective VRAC modulators are likely to be discovered soon, opening promising perspectives in the treatment of this lethal tumor. 

## Figures and Tables

**Figure 1 cancers-11-00307-f001:**
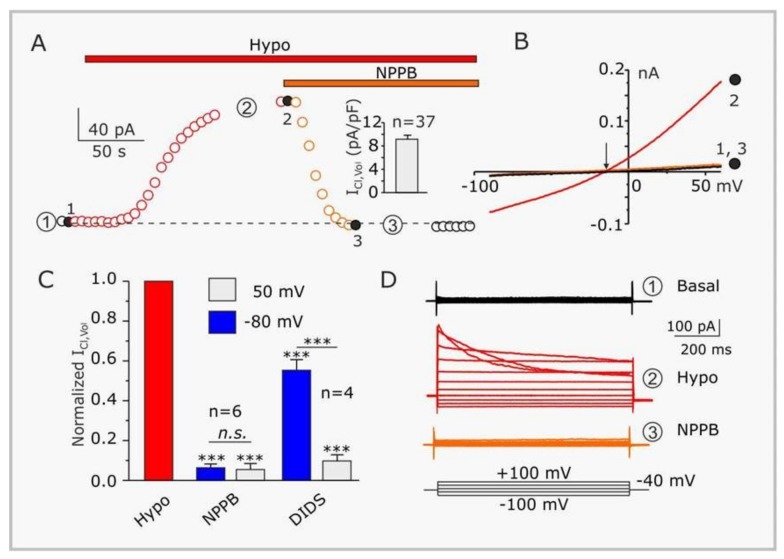
Biophysical and pharmacological properties of the hypotonicity-activated current expressed in GL-15 glioblastoma cells. (**A**) Time course of the current, activated by 30% hypotonic solution (Hypo), and subsequently blocked by NPPB (100 µM). The plot was constructed from the current ramps of the type shown in panel B). Inset: Bar plot showing the mean density of the NPPB-sensitive current assessed at +50 mV in 37 GL-15 cells. (**B**) Representative current ramps under basal (isotonic) conditions (1), in the presence of a 30% hypotonic solution (2), and in the presence of the hypotonic solution containing 100 µM NPPB (3). The arrow indicates the reversal potential of the hypotonic-activated current. (**C**) Bar plot showing the effects of 100 µM NPPB (*n* = 6) or 500 µM DIDS (*n* = 4) on the hypotonic-activated current. (**D**) Families of current traces obtained by applying to the same GL-15 cell shown in panels A) and B) 1 s voltage steps from −100 to +100 mV, in steps of 20 mV, from a holding potential of −40 mV, under basal conditions (Basal, 1), in the presence of a 30% hypotonic solution (Hypo, 2), and in the presence of a hypotonic solution containing 100 µM NPPB (NPPB, 3). Modified from Catacuzzeno et al. [4].

**Figure 2 cancers-11-00307-f002:**
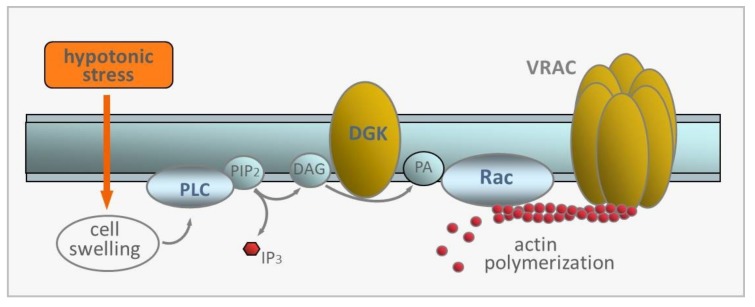
Hypothetical sequentially coupled transduction pathway leading to the activation of VRAC by hypotonicity-induced cell swelling. PLC, phospholipase C; PIP_2_, phosphatidylinositol bisphosphate; IP_3_, inositol trisphosphate; DAG, diacylglycerol; DGK, DAG kinase; PA, phosphatidic acid. Rac, small signaling G protein, member of Rho family of GTPases.

**Figure 3 cancers-11-00307-f003:**
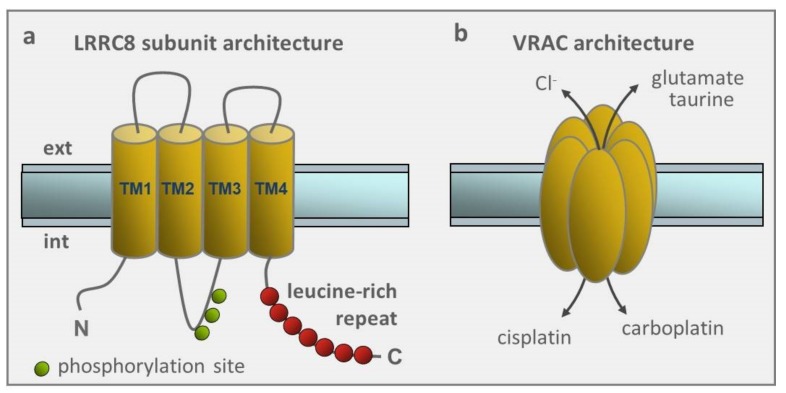
Architecture of LRRC8 subunits and VRAC. (**a**) Transmembrane topology of LRRC8 subunits proposes four transmembrane domains (TM1-4) and a C terminus with up to 17 leucine-rich repeats. LRRC8 subunits have on average multiple phosphorylation sites (indicated here for LRRC8A subunit), and several glycosilation sites (not indicated). (**b**) Hexameric stoichiometry of VRAC, based on its partial homology with pannexin/connexin channels.

**Figure 4 cancers-11-00307-f004:**
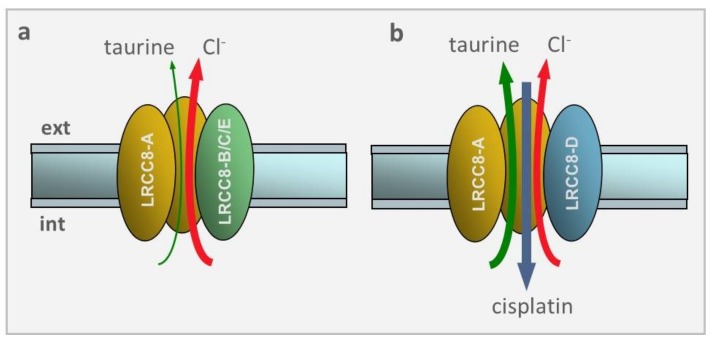
VRAC selectivity depends on the LRRC8 subunit composition. (**a**) VRAC made of the LRRC8A and LRRC8B/C/E subunits displays a large preference for Cl^−^ ions over taurine, and no permeability towards cisplatin. (**b**) VRAC made of LRRC8A and LRRC8D subunits reverses the selectivity seen before, with large preference for taurine over Cl^−^ ions. Most notably, this subunits combination, and this only one, makes VRAC very permeable to cisplatin.

**Figure 5 cancers-11-00307-f005:**
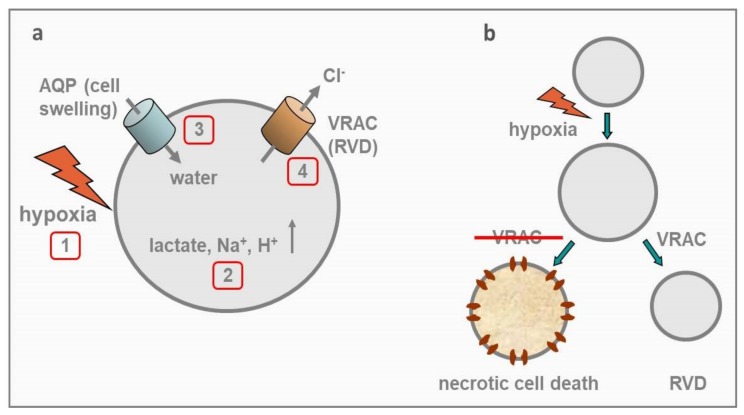
Hypoxia, VRAC, RVD, and GBM cell survival. (**a**) Sequence of events induced by hypoxia on a generic cell. Hypoxia (1) switches cell metabolism to anaerobic mode, with the result that glucose is converted to lactate, very few ATP molecules are synthesized, pump-dependent homeostasis is inhibited, and excess of Na^+^ and H^+^ follows. All these occurrences increase the cell internal tonicity (2). Obligatory osmotic water enters the cell, with consequent cell swelling (3). Cell swelling activates VRAC, which in conjunction with K channels extrude osmolites (the RVD process is activated), and cell swelling is reverted (at least in part). (**b**) Schematic illustration of the postulated role of VRAC in GBM cell survival. Following acute hypoxic insult the cell undergoes an increase in cell volume, followed by the activation of VRAC and consequent RVD process. By contrast, in the absence of VRAC the cell swelling persists, leading to necrotic cell death.

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
