# Peer review of "The Volume-Regulated Anion Channel in Glioblastoma"

_cancers, 2019, doi:10.3390/cancers11030307_

Round 1
Reviewer 1 Report
In this topical review, M. Caramia et al. summarize literature findings, which implicate volume-regulated anion channel (VRAC) into malignant properties of the deadly primary brain cancer, glioblastoma. One interesting suggestion from the Authotrs’ own work is that the enhanced activity of VRAC in glioblastoma the product of unique, cell type-specific regulation by intracellular signaling cascades involving PLC, DAG kinase, and the small GTPase rac.
I think that this manuscript has merit and will be of interest to the General Reader of Cancers as well as specialists in the field. However, there are several aspects of this work that would require additional attention. The Authors need to update their paper with proper references related to glioblastoma biology and the recent critical findings on molecular biology of VRAC. Furthermore, they need to seek an assistance of professional language editing services.
Specific comments and suggestions:
The Authors provide a solid introduction into glioblastoma biology but incorporate no references supporting their major points (with the sole exception of original paper [1]). This is unacceptable and requires additional attention. Also, the introduction does not acknowledge the current clinical classification of GBM, which segregates tumors to IHD-while type and IDH-positive. This omission may have a negative impact on credibility/perceived value of the current review in eyes of the readers with clinical background.
Understandably, the Authors place a strong emphasis on their own VRAC work. However, to make this review more balanced, I would suggest updating it with additional recent findings that are highly relevant to the presented topic. Also, in few instances, the prior literature is not accurately recapitulated or interpreted. To mention a few specific examples:
Initial statement on the ubiquitous expression of VRAC (p. 3, lns. 89-90) should be supported by reviews of K. Strange, B. Nilius, and/or Y. Okada, and the idea of functional expression of VRAC in GBM should be associated with the work of H. Sontheimer. The Authors use some at these references in the subsequent text, but they have to be introduced earlier.
There were numerous precedents in the literature suggesting that VRACs promote various aspects of malignancy. It would be helpful to see a few citations of the relevant pioneer findings in Section 2 (in addition to the Authors’ own work).
The Authors give the wrong range of the predicted sizes of VRAC pore. To the best of my recollection, studies with permeant ions suggested diameter of 6-7 nM (V.I. Ternovsky et al., 2004), and then there are more recent structural studies of LRRC8 hetero- and homomers also suggesting fairly narrow pore (e.g., J.M. Kefauver et al., eLife, 2018).
While talking about incomplete specificity of DCPIB, the Authors failed to include three important references: L. Miniery et al. (Br. J. Pharmacol., 2013) and W. Deng et al. (Pflugers Arch., 2016), both of which found the off-target blockade of several classes of K+ channels, and N. Bowens et al. (Mol. Pharmacol., 2013), who established that VRAC blocks connexin hemichannels, glial glutamate transporters, and unspecified Ca2+-activated Cl- channels, These findings suggest important limitations of the VRAC studies relying on DCPIB for their conclusions.
There are two recent important publications linking LRRC8A-containing VRAC to proliferation and chemo sensitivity in GBM cells (S. Rubino et a., Front. Oncol., 2018) and LRRC8A protein to VRAC Cl- currents in related astroglial cells (F. Formaggio et al., FASEB J., 2019). I think that they are highly relevant to the topic discussed by the Authors.
3. While talking about hypoxia, the Authors do not take into consideration a widely acknowledged phenomenon that VRAC activity requires non-hydrolytic biding of intracellular ATP. If [ATP]i drops below 20-30% of their control values, VRAC opening is largely prevented. The relevant papers from the groups of K. Strange and Y. Okada should be mentioned and discussed in this context. Also, the Authors discuss in parallel metabolic cell swelling and contribution of VRAC to RVD in metabolically inhibited cells. In the face of the Na+,K+-ATPase failure RVD cannot be sustained due to disrupted K+ gradients.
4. I recommend highly to the Authors to recruit professional language editing services to improve quality of the English in this manuscript.
Minor concerns:
I do not understand why some of the important quoted papers are mentioned in the footnotes only rather than in the main text and omitted in the list of cited literature. Please double-check.
P.3, ln. 87: VOLUME-regulated anion channels.
Reference 12 is the wrong reference of S. Pedersen. Perhaps the Authors meant to quote of her VRAC-related reviews.
Author Response
In this topical review, M. Caramia et al. summarize literature findings, which implicate volume-regulated anion channel (VRAC) into malignant properties of the deadly primary brain cancer, glioblastoma. One interesting suggestion from the Authotrs’ own work is that the enhanced activity of VRAC in glioblastoma the product of unique, cell type-specific regulation by intracellular signaling cascades involving PLC, DAG kinase, and the small GTPase rac.
I think that this manuscript has merit and will be of interest to the General Reader of Cancers as well as specialists in the field. However, there are several aspects of this work that would require additional attention. The Authors need to update their paper with proper references related to glioblastoma biology and the recent critical findings on molecular biology of VRAC. Furthermore, they need to seek an assistance of professional language editing services.
We thank the reviewer for his/her comments, which have greatly helped us to improve the Ms, including the suggestion for language revision which has been done by an English language professional.
Specific comments and suggestions:
1. The Authors provide a solid introduction into glioblastoma biology but incorporate no references supporting their major points (with the sole exception of original paper [1]). This is unacceptable and requires additional attention. Also, the introduction does not acknowledge the current clinical classification of GBM, which segregates tumors to IHD-while type and IDH-positive. This omission may have a negative impact on credibility/perceived value of the current review in eyes of the readers with clinical background.
Following the reviewer’s comments we have significantly increased the references on glioblastoma biology (l. 34-39 of revised Ms), and updated its molecular classification in IDH-wild type, IDH-mutant and NOS, as suggested by recent advances in molecular genetics and the World Health Organization in 2016 (l. 40-52).
2. Understandably, the Authors place a strong emphasis on their own VRAC work. However, to make this review more balanced, I would suggest updating it with additional recent findings that are highly relevant to the presented topic. Also, in few instances, the prior literature is not accurately recapitulated or interpreted. To mention a few specific examples:
Initial statement on the ubiquitous expression of VRAC (p. 3, lns. 89-90) should be supported by reviews of K. Strange, B. Nilius, and/or Y. Okada, and the idea of functional expression of VRAC in GBM should be associated with the work of H. Sontheimer. The Authors use some of these references in the subsequent text, but they have to be introduced earlier. There were numerous precedents in the literature suggesting that VRACs promote various aspects of malignancy. It would be helpful to see a few citations of the relevant pioneer findings in Section 2 (in addition to the Authors’ own work).
In the section where we describe the general properties of VRAC, we have significantly increased the number of references on VRAC general expression (i.e Strange et al., 1996; Nilius et al., 1997; Jentsch, 2016) and its expression in glioblastoma (Cuddapah and Sontheimer, 2011). Additional references have been added when, with regard to our own work, we describe the role of VRAC in glioblastoma malignancy (l. 245-250).
3. The Authors give the wrong range of the predicted sizes of VRAC pore. To the best of my recollection, studies with permeant ions suggested diameter of 6-7 nM (V.I. Ternovsky et al., 2004), and then there are more recent structural studies of LRRC8 hetero- and homomers also suggesting fairly narrow pore (e.g., J.M. Kefauver et al., eLife, 2018).
Ternovsky et al., 2004 made indeed an accurate estimate of VRAC pore size and found a polymer hydrodynamic radius cut-off of 0.63 nm, in line with all previous studies, namely those using calixarenes that suggested a lower and upper limits of 1.1x1.2 and 1.7x1.2 nm, respectively, for the cross-sectional dimensions (Droogmans et al., 1998, 1999) which means a minimum and maximum pore radius calculated as a geometric mean of these dimensions as 0.57 and 0.71 nm, respectively. All these studies give a converging estimate of the VRAC pore radius of 0.6–0.7 nm, which is fully consistent with the data we reported in our Ms, only considering that we used the pore diameter instead of the pore radius as reference, and Angstrom instead of nm as unit of measure.
The reviewer recalls in addition more recent structural studies of LRRC8 hetero- and homomers (e.g., Kefauver et al., eLife, 2018) apparently indicating a pore size at the selectivity filter significantly lower than previously estimated. With regard to these newer studies we added a short paragraph saying that the small pore radii reported (<3 A) are not really surprising for these homoexameric LCCR8A channels, but somehow expected from the absence of LRRC8D subunits which have been shown to form wider pores and confer broader substrate specificity (cf. Planells-Cases et al., 2015; see also Voets et al., 2015) (l. 112-116).”
4. While talking about incomplete specificity of DCPIB, the Authors failed to include three important references: L. Miniery et al. (Br. J. Pharmacol., 2013) and W. Deng et al. (Pflugers Arch., 2016), both of which found the off-target blockade of several classes of K+ channels, and N. Bowens et al. (Mol. Pharmacol., 2013), who established that VRAC blocks connexin hemichannels, glial glutamate transporters, and unspecified Ca2+-activated Cl- channels, These findings suggest important limitations of the VRAC studies relying on DCPIB for their conclusions.
Following the reviewer’s suggestions, we added few lines and the indicated references to report the off-target activity of DCPIB as activator/blocker of TREK and Kir K channels (l. 141-143).
5. There are two recent important publications linking LRRC8A-containing VRAC to proliferation and chemo sensitivity in GBM cells (S. Rubino et a., Front. Oncol., 2018) and LRRC8A protein to VRAC Cl- currents in related astroglial cells (F. Formaggio et al., FASEB J., 2019). I think that they are highly relevant to the topic discussed by the Authors.
As suggested by the reviewer, we introduced recent publications linking LRRC8-containing VRAC to glioblastoma cell proliferation and chemo-sensitivity (Rubino et al., 2018) and VRAC Cl currents in astroglial cells (Formaggio et al., 2019) (l. 90,91).
6. While talking about hypoxia, the Authors do not take into consideration a widely acknowledged phenomenon that VRAC activity requires non-hydrolytic biding of intracellular ATP. If [ATP]i drops below 20-30% of their control values, VRAC opening is largely prevented. The relevant papers from the groups of K. Strange and Y. Okada should be mentioned and discussed in this context. Also, the Authors discuss in parallel metabolic cell swelling and contribution of VRAC to RVD in metabolically inhibited cells. In the face of the Na+,K+-ATPase failure RVD cannot be sustained due to disrupted K+ gradients.
As suggested by the reviewer, in the revised Ms we now discuss the intracellular ATP dependence of VRAC activation observed in many cell preparations. Notably, this ATP dependence of VRAC does not appear to occur in glioblastoma cells, given that VRAC currents could be activated upon cell swelling also in the absence of intracellular ATP, and addition of intracellular ATP (2 mM) did not appreciably affect VRAC current (see Ransom et al., J Neurosci 2001). This finding appears to agree with our unpublished observations that in glioblastoma cells VRAC is strongly activated by hypoxia, a condition that leads to metabolic inhibition and ATP depletion (Jeong et al., Neurochem Res 2003). In the revised Ms we also discuss the occurrence that the RVD process under hypoxic conditions appears to be attenuated because reduced K gradient resulting from ATP depletion (l. 399-408).
7. I recommend highly to the Authors to recruit professional language editing services to improve quality of the English in this manuscript.
As we said earlier, we have revised the English of our Ms by recruiting professional editing service. Please notice that, because of the many small changes introduced we chose not to report them in the revised Ms. Of course they will be promptly provided upon request.
Minor concerns:
I do not understand why some of the important quoted papers are mentioned in the footnotes only rather than in the main text and omitted in the list of cited literature. Please double-check.
We double-checked text and footnotes and updated both the main text and the list of references.
P.3, ln. 87: VOLUME-regulated anion channels.
Fixed!
Reference 12 is the wrong reference of S. Pedersen. Perhaps the Authors meant to quote of her VRAC-related reviews.
We corrected the reference.
Reviewer 2 Report
This manuscript by Caramia et al. aims to review the role of volume-regulated anion channels (VRAC) in glioblastoma (GBM), the most frequent and lethal primary brain cancer in adults. The authors started by introducing GBM biology, the biophysical properties, pharmacology, and activation mechanisms of VRAC. They described the recent identification of LRRC8 isoforms as the molecular components of VRAC. They stated the role of VRAC in regulating cell volume and transporting of chemotherapeutic agents. Lastly, the authors described two potential mechanisms that may regulate VRAC activity in GBM: the presence of blood serum from the leaky tumor vessel and the hypoxic microenvironment. Targeting ion channels in cancer is an emerging and important field. Therefore, the authors wrote a timely review on an intriguing topic by focusing on a class of ion channels that are relatively unexplored. Furthermore, this review is composed of multiple sections with a logic flow that should facilitate the audience’s reading and interpretations. Despite these laudable aspects, unfortunately the review lacks sufficient details to clearly demonstrate its points at various points, certain experimental caveats and limitations are not described or discussed, and some statements appears to be biased or not supported by current views. Moreover, the title of the review does not fit with its text. My specific comments are listed below. 1. This review does not spend much effort on discussing how VRAC enhances GBM malignancy. Rather, it reads like a general (and somewhat superficial) introduction of GBM, VRAC, and how VRAC may be activated in GBM. Therefore, the title does not fit with the content. 2. The authors used the term VRAC throughout the main text. To be consistent, I would suggest using volume regulated anion channel, instead of swelling-activated Cl current, in their title. 3. In the Abstract, the authors claimed that “volume-regulated anion channels (VRAC), which mediates the swelling-activated chloride current, IClswell, and is highly expressed in GBM cells, plays by far the dominant role (than other types of ion channels)”. Why? What is the evidence that VRAC is more functionally important and/or relevant than other ion channels in GBM? 4. Related to point 3, there are many published reports on the role of CLCs and CLIC channels in GBM. The authors should cite these references and discuss the significance of those findings in the context of describing the role of VRACs. 5. The authors mentioned that VRAC is highly expressed in GBM cells. Please provide the references for this statement. Given the high level of heterogeneity in GBM, what is the expression pattern of VRAC in the various GBM subtypes? Is high VRAC expression correlated with the alterations of specific signaling pathways or certain genetic mutations? GBM consists of a hierarchy of cell types. Is VRAC expressed in the cancer stem cells, transiently amplifying progenitor cells, or post-mitotic cells? It is important to note that serum-cultured cells are very poor model of in vivo tumor biology. 6. Page 2, line 55: the non-tumoral brain tissue surrounding GBM is not necessarily “healthy”. There are active tumor-stromal interactions that can alter the brain parenchyma. “non-tumor brain tissue” is a more accurate description. 7. Page 2, line 61: it seems biased to state that “migration of tumor cells is often viewed as result of mutation-induced inhibition of specific biochemical pathways that in healthy tissues keeps down cell migration”. There are numerous examples of GBM cell migration due to activation of certain signaling mechanisms. 8. Page 3, line 87: volume-regulated anion channel” not “voltage”. 9. The authors’ finding and discussion that the PLC/DAG/DGK/PA pathway and Rac-mediated cytoskeletal remodeling pathway may regulate VRAC function is interesting. Do they have any hypothesis regarding how actin remodeling impact on VRAC? 10. First paragraph of page 7: the lack of phenotype in LRRC8 cells may well be that hitherto unidentified proteins function as VRAC. The authors need to include this possibility in the discussion. 11. The authors mentioned that VRAC-mediated cell volume regulation is essential for cell proliferation, migration and death. It is necessary to clearly describe how cell volume dynamics is important for these various biological processes, so that the readers can appreciate the significance of VRAC in regulating volume. 12. From page 9 to 12 the authors described serum and hypoxia as two possible mechanisms in GBM that may activate VRAC. However, these descriptions do not provide insights into how VRAC activity enhances the aggressive phenotype of GBM, as the section title or the manuscript title reads. The authors need to significantly revise or re-write the text if what they really want to discuss is how VRAC regulates GBM biology. As stated above, serum-cultured cells are poor model for in vivo tumors. If not much is known about the in vivo role of VRAC in GBM tumors in animals or humans, it would be more accurate to state that what’s known and reviewed in the manuscript is VRAC function in GBM cells, not tumors. 13. First paragraph of page 12: how specific is DCPIB? Has the claim that VRAC mediates RVD under hypoxic condition in GBM been confirmed by studying the role of LRRC8? This needs to be discussed. 14. Second paragraph of page 12: “We have shown that VRAC are widely expressed in GBM cells” is an over-statement. How many different GBM molecular subtypes, GBM cell types along the cellular hierarchy, GBMs of different genetic mutations etc. have been studied to reach to this conclusion?
Author Response
This manuscript by Caramia et al. aims to review the role of volume-regulated anion channels (VRAC) in glioblastoma (GBM), the most frequent and lethal primary brain cancer in adults. The authors started by introducing GBM biology, the biophysical properties, pharmacology, and activation mechanisms of VRAC. They described the recent identification of LRRC8 isoforms as the molecular components of VRAC. They stated the role of VRAC in regulating cell volume and transporting of chemotherapeutic agents. Lastly, the authors described two potential mechanisms that may regulate VRAC activity in GBM: the presence of blood serum from the leaky tumor vessel and the hypoxic microenvironment. Targeting ion channels in cancer is an emerging and important field. Therefore, the authors wrote a timely review on an intriguing topic by focusing on a class of ion channels that are relatively unexplored. Furthermore, this review is composed of multiple sections with a logic flow that should facilitate the audience’s reading and interpretations. Despite these laudable aspects, unfortunately the review lacks sufficient details to clearly demonstrate its points at various points, certain experimental caveats and limitations are not described or discussed, and some statements appears to be biased or not supported by current views. Moreover, the title of the review does not fit with its text. My specific comments are listed below.
1. This review does not spend much effort on discussing how VRAC enhances GBM malignancy. Rather, it reads like a general (and somewhat superficial) introduction of GBM, VRAC, and how VRAC may be activated in GBM. Therefore, the title does not fit with the content.
2. The authors used the term VRAC throughout the main text. To be consistent, I would suggest using volume regulated anion channel, instead of swelling-activated Cl current, in their title.
Following the reviewer’s comments, we changed the title of the Ms, which now reads “The volume-regulated anion channel in glioblastoma cells”.
3. In the Abstract, the authors claimed that “volume-regulated anion channels (VRAC), which mediates the swelling-activated chloride current, IClswell, and is highly expressed in GBM cells, plays by far the dominant role (than other types of ion channels)”. Why? What is the evidence that VRAC is more functionally important and/or relevant than other ion channels in GBM?
We changed “dominant” with “major”, removed the compelling “by far”, and slightly rephrased the sentence to soften the statement (l. 15-17). In any case, with the term “dominant” we meant to underline the critical importance played by VRAC in sustaining, in conjunction with K channels, the efflux of KCl needed for volume changes that take place in several processes of glioblastoma aggressiveness.
4. Related to point 3, there are many published reports on the role of CLCs and CLIC channels in GBM. The authors should cite these references and discuss the significance of those findings in the context of describing the role of VRACs.
These references were not included in our review in the light of the recent discovery of LRCC8A-E as the constituents of VRAC, and the lack of stringent evidence that ClC and CLIC channels are activated by volume increase.
5. The authors mentioned that VRAC is highly expressed in GBM cells. Please provide the references for this statement. Given the high level of heterogeneity in GBM, what is the expression pattern of VRAC in the various GBM subtypes? Is high VRAC expression correlated with the alterations of specific signaling pathways or certain genetic mutations? GBM consists of a hierarchy of cell types. Is VRAC expressed in the cancer stem cells, transiently amplifying progenitor cells, or post-mitotic cells? It is important to note that serum-cultured cells are very poor model of in vivo tumor biology.
Several references on VRAC expression in GBM cells are now given (l. 89- 91). With regard to the other point, to our knowledge virtually no information on VRAC expression in the various types of glioblastoma is presently available. This shortage is mainly due to the lack of a molecular counterpart for VRAC that we have had for more than 25 years from the first report of VRAC. Hopefully, the identification of LRRC8 as structural component of VRAC, reported in 2014, will rapidly fill this gap.
6. Page 2, line 55: the non-tumoral brain tissue surrounding GBM is not necessarily “healthy”. There are active tumor-stromal interactions that can alter the brain parenchyma. “non-tumor brain tissue” is a more accurate description.
We changed “ healthy brain parenchyma” into “non-tumour brain tissue”.
7. Page 2, line 61: it seems biased to state that “migration of tumor cells is often viewed as result of mutation-induced inhibition of specific biochemical pathways that in healthy tissues keeps down cell migration”. There are numerous examples of GBM cell migration due to activation of certain signaling mechanisms.
We added some examples of signal transduction pathways involved in glioblastoma cell migration (cf l.62-63).
8. Page 3, line 87: volume-regulated anion channel” not “voltage”.
Fixed!
9. The authors’ finding and discussion that the PLC/DAG/DGK/PA pathway and Rac-mediated cytoskeletal remodeling pathway may regulate VRAC function is interesting. Do they have any hypothesis regarding how actin remodeling impact on VRAC?
Unfortunately we don’t. To our knowledge there hasn’t been any reported modulation of VRAC by cytoskeleton remodeling so far.
10. First paragraph of page 7: the lack of phenotype in LRRC8 cells may well be that hitherto unidentified proteins function as VRAC. The authors need to include this possibility in the discussion.
As suggested by the reviewer, we added the sentence “Finally, the possibility is still open that VRAC expressed in some cells may have other still unidentified molecular counterparts.” (l. 224-225)
11. The authors mentioned that VRAC-mediated cell volume regulation is essential for cell proliferation, migration and death. It is necessary to clearly describe how cell volume dynamics is important for these various biological processes, so that the readers can appreciate the significance of VRAC in regulating volume.
We have now stressed the importance of cell volume changes in these cellular processes (l. 245-250)
12. From page 9 to 12 the authors described serum and hypoxia as two possible mechanisms in GBM that may activate VRAC. However, these descriptions do not provide insights into how VRAC activity enhances the aggressive phenotype of GBM, as the section title or the manuscript title reads. The authors need to significantly revise or re-write the text if what they really want to discuss is how VRAC regulates GBM biology. As stated above, serum-cultured cells are poor model for in vivo tumors. If not much is known about the in vivo role of VRAC in GBM tumors in animals or humans, it would be more accurate to state that what’s known and reviewed in the manuscript is VRAC function in GBM cells, not tumors.
In accordance with the reviewer’s comments, we changed the section title from “VRAC activity enhances the aggressive phenotype of GBM” to “Activation of VRAC by pathologically relevant conditions”. In addition, in closing up our review we added a sentence stating that it remains to be seen whether our observations in GBM cell lines also apply to GBM cells in situ (l. 417-418)
13. First paragraph of page 12: how specific is DCPIB? Has the claim that VRAC mediates RVD under hypoxic condition in GBM been confirmed by studying the role of LRRC8? This needs to be discussed.
One of the relevant limits in studying VRAC has been the lack of selective pharmacological blockers. DCPIB is undoubtedly the most effective, yet it also mediates off-target blockade or activation of inward rectifier and TREK K channels, respectively. In the revised version of the Ms we’ve now discussed the unspecificity of DCPIB (l. 141-143). It needs to be added, though, that it has been used in several studies (cf for instance Wong et al., J Cell Physiol 2018) as an effective tool to study VRAC impact on GBM cell migration, proliferation and invasion. To our knowledge, no information about the putative hypoxic-mediated LRRC8 modulation is presently available.
14. Second paragraph of page 12: “We have shown that VRAC are widely expressed in GBM cells” is an over-statement. How many different GBM molecular subtypes, GBM cell types along the cellular hierarchy, GBMs of different genetic mutations etc. have been studied to reach to this conclusion?
Following the reviewer’s comments we removed the sentence indicated which in our view would convey the message that we routinely recorded markedly high levels of VRAC current density in every cell line we used in our studies.
Reviewer 3 Report
The authors present a comprehensive view on the role of volume-regulated anion channels in the metastasis of glioblastoma multiform. It is an exciting, informative, and timely review.
I have a few minor comments.
1) Line 120: ...'‘hypotonic-activated current’ should be ‘hypotonicity-activated current’
2) Please refer to the individual panels of the figures in the text (all Figure 1 panels and Figure 5b are missing in the text).
3) Figure 3 is not mentioned at all in the main text. Please refer to it, including its panels.
4) The following sentence in the legend of Figure 4 is not comprehensible.:’ Most notable, this subunits combination, and only this, makes VRAC very permeable to 268 cisplatin.’ Please revise the sentence.
5) line 287: “in vitro’ should be italic
Author Response
Comments and Suggestions for Authors
The authors present a comprehensive view on the role of volume-regulated anion channels in the metastasis of glioblastoma multiform. It is an exciting, informative, and timely review.
I have a few minor comments.
1) Line 120: ...'‘hypotonic-activated current’ should be ‘hypotonicity-activated current’
We changed the sentence as suggested by the reviewer.
2) Please refer to the individual panels of the figures in the text (all Figure 1 panels and Figure 5b are missing in the text).
We added in the text explicit reference to each individual panel of the figures.
3) Figure 3 is not mentioned at all in the main text. Please refer to it, including its panels.
We added comments to Figure 3 in the main text, and to each panel.
4) The following sentence in the legend of Figure 4 is not comprehensible.:’ Most notable, this subunits combination, and only this, makes VRAC very permeable to 268 cisplatin.’ Please revise the sentence.
We recruited professional english language editing and then the sentence in the Figure 4 has been modified.
5) line 287: “in vitro’ should be italic
Done!
Round 2
Reviewer 2 Report
The revision by the authors is appreciated. Two issues remain.
1. Line 61-63: The statement "the migration of tumor cells is often viewed as the result of mutation-induced inhibition of specific biochemical pathways as, for example, PI3K/Akt or mTor that in healthy tissues keep down the migration of cells" is vague and inaccurately simplified. The authors need to provide specific examples by citing specific papers to support which mutation results in inhibition (or activation) or which signaling pathway, which promotes tumor cell migration (and under what circumstance the same pathway inhibits cell migration in heathy tissues.
2. Line 247: Cell volume undergoes dynamic alterations (oscillatory volume increases and decreases) during cell cycle. Perturbation of cell volume in either direction is detrimental for cell cycle progression. Therefore, the statement that "For example cell proliferation is stimulated by cell swelling and inhibited by cell shrinkage" is incorrect. The authors need to revise this statement and cite multiple references to reflect this current understanding in the field.
Author Response
1. Line 61-63: The statement "the migration of tumor cells is often viewed as the result of mutation-induced inhibition of specific biochemical pathways as, for example, PI3K/Akt or mTor that in healthy tissues keep down the migration of cells" is vague and inaccurately simplified. The authors need to provide specific examples by citing specific papers to support which mutation results in inhibition (or activation) or which signaling pathway, which promotes tumor cell migration (and under what circumstance the same pathway inhibits cell migration in heathy tissues.
We thank again the reviewer for his/her further effort in helping us to improve our Ms.
Following the reviewer´s comment, we reworded the indicated sentence by replacing “mutation-induced inhibition” with the more appropriate “dysregulation”. We then discussed more accurately one of the most important signal transduction pathways – the PI3K/Akt/Rho – whose dysregulation plays a central role in the aberrant glioblastoma cell migration/invasion, and added the appropriate references.
Finally, on reading this reorganized part we found more appropriate to move the last five lines of the original section ( “Cell migration … sorrounding parenchyma”; originally, l. 60-65) few lines above (to l. 58-63) to chain it with another sentence addressing the same aspect.
2. Line 247: Cell volume undergoes dynamic alterations (oscillatory volume increases and decreases) during cell cycle. Perturbation of cell volume in either direction is detrimental for cell cycle progression. Therefore, the statement that "For example cell proliferation is stimulated by cell swelling and inhibited by cell shrinkage" is incorrect. The authors need to revise this statement and cite multiple references to reflect this current understanding in the field.
We agree with the reviewer’s comment that during the cell cycle cell volume undergoes significant dynamic changes. However, to our knowledge, cell volume changes induced by altering the osmolarity of the medium correlate with proliferation the way we have reported in the Ms. In the revised Ms we have therefore only added some more references to better support the point made (l. 256).